# Transient drug-tolerance and permanent drug-resistance rely on the trehalose-catalytic shift in *Mycobacterium tuberculosis*

Jae Jin Lee [1,6], Sun-Kyung Lee[2,6], Naomi Song[3], Temitope O. Nathan[4], Benjamin M. Swarts[4], Seok-Yong Eum[2], Sabine Ehrt[3], Sang-Nae Cho[2,5] & Hyungjin Eoh [1]

Stochastic formation of *Mycobacterium tuberculosis* (Mtb) persisters achieves a high level of antibiotic-tolerance and serves as a source of multidrug-resistant (MDR) mutations. As conventional treatment is not effective against infections by persisters and MDR-Mtb, novel therapeutics are needed. Several approaches were proposed to kill persisters by altering their metabolism, obviating the need to target active processes. Here, we adapted a biofilm culture to model Mtb persister-like bacilli (PLB) and demonstrated that PLB underwent trehalose metabolism remodeling. PLB use trehalose as an internal carbon to biosynthesize central carbon metabolism intermediates instead of cell surface glycolipids, thus maintaining levels of ATP and antioxidants. Similar changes were identified in Mtb following antibiotic-treatment, and MDR-Mtb as mechanisms to circumvent antibiotic effects. This suggests that trehalose metabolism is associated not only with transient drug-tolerance but also permanent drug-resistance, and serves as a source of adjunctive therapeutic options, potentiating antibiotic efficacy by interfering with adaptive strategies.

---

[1] Department of Molecular Microbiology and Immunology, Keck School of Medicine, University of Southern California, Los Angeles, CA 90033, USA.
[2] Division of Immunopathology and Cellular Immunology, International Tuberculosis Research Center, Changwon 51755, Republic of Korea. [3] Department of Microbiology and Immunology, Weill Cornell Medical College, New York, NY 10065, USA. [4] Department of Chemistry and Biochemistry, Central Michigan University, Mount Pleasant, MI 48859, USA. [5] Department of Microbiology and Institute of Immunology and Immunological Disease, Yonsei University College of Medicine, Seoul 03722, Republic of Korea. [6]These authors contributed equally: Jae Jin Lee, Sun-Kyung Lee. Correspondence and requests for materials should be addressed to H.E. (email: heoh@usc.edu)

Tuberculosis (TB) has afflicted humans since the Neolithic age and is still the 10th leading cause of human death worldwide in the 21st century[1]. Conventional TB treatment consists of a 2-month intensive phase followed by a 4-month continuation phase[2,3]. The efficacy of conventional TB treatment is blunted largely due to the lengthy treatment with a highly intensive drug regimen and the ability of *Mycobacterium tuberculosis* (Mtb)—the causative agent of TB—to form persister cells (persisters), a small fraction of phenotypic variants that are tolerant to nearly all TB drugs[4–7]. During its decades-long interactions with growth-adverse environments, Mtb has evolved to overcome these stresses by stochastic formation of persisters and accompanied adaptive strategies[8–11]. Thus, Mtb persisters constitute a therapeutically critical but largely understudied facet of the TB pandemic.

Antibiotic modes of action were typically thought to be determined by both the drug-to-target interaction and the resulting cellular consequences on target bacteria[12–14]. However, a recent report proposed the production of deleterious reactive oxygen species (ROS) by antibiotics as a common bactericidal effector[15–18]. A recent metabolomics study using Mtb following single treatment with various first-line TB drugs showed that Mtb is intrinsically drug tolerant because of its ability to remodel metabolic networks and electron transport chain (ETC) activities in order to circumvent antibiotic-mediated ROS production and irreversible antimicrobial damage[19]. Also, accumulating evidence showed that dysregulated ETC activities and ATP depletion are two molecular signals that enhance bacterial persister formation[20–23]. Thus, we hypothesize that Mtb persisters monitor the metabolic vulnerability induced by ETC dysregulation and ATP depletion following antibiotic treatment, and respond by modifying the perturbed metabolic networks to gain drug tolerance[24].

Trehalose is an abundant nonreducing glucose disaccharide in Mtb[25,26]. Thus, Mtb trehalose serves as both a carbohydrate store as well as a core component of cell surface trehalose monomycolate (TMM) and trehalose dimycolate (TDM)[26,27]. A recent lipidomics study has shown that Mtb enters a non-replicating drug-tolerant state in response to hypoxia by downregulating TMM/TDM and remodeling trehalose metabolism[28–30], implying active involvement of trehalose in the adaptive processes for emergence of Mtb drug tolerance.

A major barrier in studying Mtb persister physiology has been the lack of an optimal in vitro culture system and feasible analytic tools. Here, we adapted the in vitro mycobacterial biofilm culture to model drug-tolerant persisters[31–33]. Using the isotopic metabolomics profile of the persister subfraction, we observed that a catalytic shift of trehalose metabolism keeps trehalose and maltose away from the biosynthesis of cell wall TMM and TDM and channels trehalose and maltose into the biosynthesis of central carbon metabolism (CCM) intermediates in order to maintain ATP and antioxidant biosynthetic activities. Intriguingly, this trehalose-catalytic shift was also active in drug-resistant, but surprisingly not of drug-sensitive TB clinical isolates under an antibiotic-free condition. Thus, the trehalose-catalytic shift is an essential component of adaptive metabolism that not only permits survival in response to a dysregulated ETC and ATP depletion, but also facilitates—either directly or indirectly—the accumulation of drug-resistant mutations in Mtb.

## Results

**Mtb biofilm culture is enriched with drug-tolerant PLB.** We adapted a mycobacterial biofilm culture system[32] to model the in vitro Mtb culture that is enriched with persister-like bacilli (PLB). This biofilm system allowed us to reproducibly generate Mtb PLB in pelleted form at both the wall-media border and air-media interface (Supplementary Fig. 1a). Floating and precipitated fractions, which contain replicating and dead bacilli, respectively, were removed to selectively collect PLB-enriched Mtb culture[34]. A 5-week-long incubation in the adapted biofilm system permitted gradual enrichment with PLB through three stages: initiation (days 0–16), accretion (days 16–28), and maturation (day 28 and after), which were defined by the slope of crystal violet (CV) staining kinetics, tolerance against first-line TB drugs isoniazid (INH) and rifampicin (RIF), and mRNA expression kinetics of *lat*, a validated transcriptional marker of Mtb persisters[7,35] (Supplementary Fig. 1b–e). PLB enrichment was inversely related to levels of external carbon consumption and ATP biosynthesis (Supplementary Fig. 1f, g). To validate that PLB represents Mtb persisters that formed following drug treatment, we generated Mtb persisters by treatment with 100 μg mL$^{-1}$ d-cycloserine (DCS) for 14 days[9]; mRNA profiles of 54 genes (Supplementary Table 1) were compared with those of PLB collected at days 16, 22, and 28 of biofilm culture (Supplementary Fig. 2a). Comparative mRNA profiling revealed that 28-day-old PLB displayed highly similar transcriptional responses to those of DCS-mediated Mtb persisters (Supplementary Fig. 2b). This suggested that acquisition of transient drug tolerance is largely attributed to adaptive remodeling of cellular metabolic activities, and the in vitro biofilm system is an experimentally feasible method for further metabolomics characterization.

**Trehalose metabolism is altered in PLB.** To identify the contribution of adaptive metabolism to transient drug tolerance of PLB, we characterized a time course of PLB metabolomics profiles. Removal of replicating and dead fractions helped achieve the signals associated with PLB-specific metabolism without dilution by unwanted noise (Supplementary Fig. 1a). Metabolic remodeling associated with PLB formation was determined by comparing the relative abundance of ~260 Mtb metabolites (Supplementary Fig. 3a). Using MetaboAnalyst (ver 4.0), multivariate hierarchical clustering analyses identified a subset of metabolites unique to PLB and mapped Mtb-annotated pathways (Supplementary Fig. 3). Consistent with mRNA profiling (Supplementary Fig. 2), the CV curve, and drug-sensitivity kinetics (Supplementary Fig. 1b–d), the results of a clustered heatmap and principal component analysis indicated that the metabolic alterations that occurred in PLB at and after day 28 were most likely associated with metabolic efforts to acquire drug tolerance (Supplementary Fig. 3a, b). A pathway impact score and log [P] of MetaboAnalyst results indicated trehalose metabolism as a top-ranked pathway, and CCM including glycolysis (GL) and the pentose phosphate pathway (PPP) as those belonging to high-ranked pathways (Supplementary Fig. 3c, d). The initial GL intermediate, glucose phosphate (glucose-P), is catalytically linked to both trehalose metabolism and PPP. Thus, we hypothesize that the alterations at the interplay between trehalose metabolism and GL/PPP play a crucial role in PLB formation and accompanied drug tolerance.

**Trehalose fuels the central carbon metabolism of PLB.** In Mtb, trehalose has multiple cellular roles, including carbohydrate storage and a core component of glycolipids. To accommodate its diverse functional repertoire, Mtb has multiple trehalose metabolism machineries; CmrA, OtsA/B, Rv2402, TreS, and Pep2 (Supplementary Fig. 4)[26]. A time-course metabolomics profile of PLB showed gradual depletion of trehalose at and after day 28 with reciprocal induction of maltose, implying specific upregulation of TreS-mediated trehalose-to-maltose conversion activity[36] (Fig. 1a, b, Supplementary Fig. 4). The levels of TMM/TDM, UDP glucose, and maltose phosphate (maltose-P) all decreased at the same time point (Fig. 1a–c). Decreased TMM/TDM with

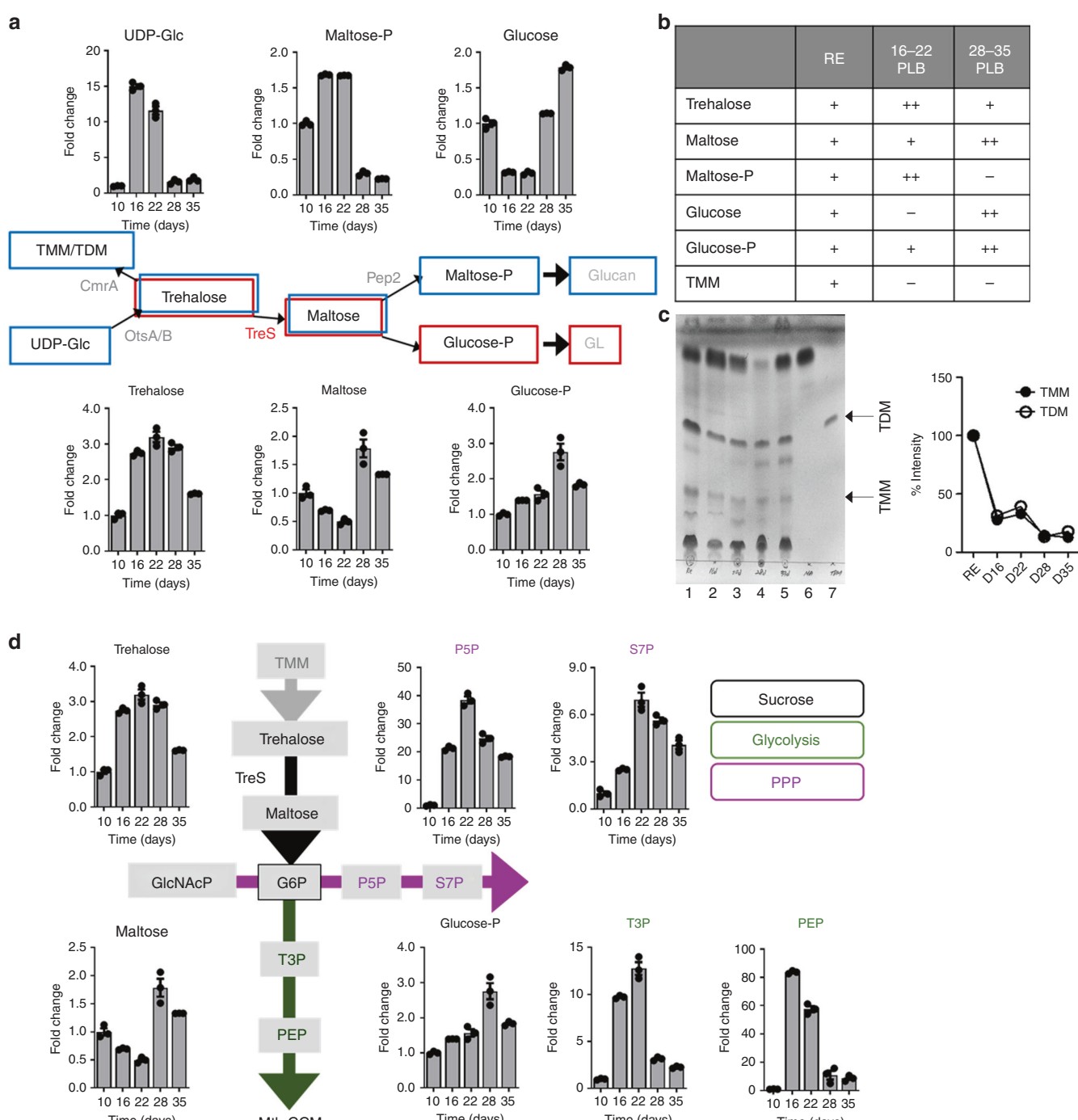

**Fig. 1** Mtb PLB-specific remodeling of trehalose metabolism, glycolysis, and the pentose phosphate pathway. **a, d** Intermediates of trehalose metabolism were as described in Supplementary Fig. 4. Intrabacterial pool sizes of Mtb intermediates in trehalose metabolism, glycolysis, and the pentose phosphate pathway in PLB harvested from mycobacterial biofilm culture at days 10, 16, 22, 28, and 35. Total bar heights indicate the intrabacterial pool sizes relative to those of PLB at day 10. Blue-boxed metabolites denote the carbon flux of the replicating state and red-boxed metabolites denote the carbon flux of PLB at day 28. UDP-Glc, uracil diphosphate-glucose; maltose-P, maltose phosphate; P5P, pentose 5-phosphates; S7P, sedoheptulose 7-phosphates; G6P, glucose 6-phosphates; T3P, triose 3-phosphates (dihydroxyacetone phosphate and glyceraldehyde phosphate); PEP, phosphoenolpyruvate. **b** Major changes in time-course PLB metabolomics (one plus, levels in replicating state; two pluses, increased; one minus, decreased). **c** Analysis of the extractable trehalose monomycolate (TMM), trehalose dimycolate (TDM), and free mycolic acids from PLB at the same time points as metabolome extraction. Equal amounts of total cellular lipids were run in the solvent system (chloroform:methanol:$H_2O$, 90:10:1; v/v/v) (left panel) and TMM/TDM intensity (density) was monitored by ImageJ software (right panel). The labeled band shown in the TLC corresponds to the $R_f$ of a TMM, TDM, or mycolic acid standard. Lane 1, replicating; Lane 2, day 16 PLB; Lane 3, day 22 PLB; Lane 4, day 28 PLB; Lane 5, day 35 PLB; Lane 6, standard mycolic acid; Lane 7, standard TDM. **d** Black font metabolites are members of sucrose metabolism, green font metabolites are members of glycolysis, and magenta font metabolites are members of the pentose phosphate pathway (PPP). CCM, central carbon metabolism. All values are the average of two independent biological duplicates ± s.e.m. Source data are provided as a Source Data file

induced maltose indicated that TreS outcompeted CmrA in catalyzing trehalose as a substrate. We also observed glucose-P induction that was kinetically matched to reciprocal depletion in Pep2-mediated maltose-P biosynthesis (Fig. 1a, Supplementary Fig. 4), implying that maltose was a preferred substrate for glucose-P, an initial activity for the biosynthesis of GL and PPP intermediates instead of α-glucan biosynthesis. Next, glucose-P, levels of PPP intermediates, and lower GL intermediates in PLB were kinetically matched (Fig. 1d). Unlike PPP intermediates, GL intermediates, including dihydroxyacetone phosphate and glyceraldehyde phosphate (collectively referred to as T3P) and phosphoenolpyruvate (PEP) were reduced after day 28, but still remained higher than they were at the initiation stage. The foregoing metabolomics profiles suggested that trehalose serves as the carbon substrate for the biosynthesis of GL and the PPP intermediates (Fig. 1d).

**TreS is responsible for the trehalose-catalytic shift in PLB**. Consistent with the PLB metabolomics profiles (Fig. 1), we observed specific upregulation of the *treS* mRNA transcript with reciprocal downregulation of other genes in trehalose metabolism (Supplementary Figs. 4 and 5a). To test if TreS is essential for PLB formation, we used *treS*-deficient Mtb (ΔtreS)[28] and observed that ΔtreS generated PLB only after a significant delay, while *otsB1*-deficient Mtb (ΔotsB1) and *rv2402*-deficient Mtb (Δrv2402) were capable of forming intact PLB at levels comparable with wild type (WT; Supplementary Fig. 5b), despite no discernible defect in planktonic growth in media containing carbohydrate or acetate (Supplementary Fig. 5c). An essential role of TreS in PLB formation was also true in persister formation. We showed that a TreS deficiency resulted in a significant reduction in DCS-mediated persister formation by an ~2 log₁₀ CFU in ΔtreS, with a greater reduction in CFU during days 7–14 as compared with that of WT Mtb (Supplementary Fig. 5d). Notably, the Mtb formed persister-enriched population following DCS treatment by using TreS-mediated metabolic remodeling. Due to the severely compromised ability of ΔtreS to form PLB, we collected multiple wells to conduct metabolomics profiling. Metabolomics profiles of ΔtreS PLB showed gradual accumulation of trehalose with no maltose production (Fig. 2a). Defective maltose production was accompanied by a failure to accumulate glucose-P and intermediates of GL and the PPP, albeit GL and the PPP intermediates of ΔtreS were partly restored only after a significant delay (Fig. 2a). Intriguingly, the ΔtreS PLB failed to downregulate TMM/TDM; instead, there was an accumulation of TMM/TDM due to the absence of the trehalose-catalytic shift (Fig. 2b), indicating that the shift relies not only on the free form of trehalose but also on trehalose released from dissociated TMM/TDM[37]. Of note, the metabolomics profiles of ΔotsB1 and Δrv2402 PLB at day 28 confirmed a specific role of TreS in both the trehalose-catalytic shift and subsequent PLB formation (Supplementary Fig. 5b, e). We also examined the mRNA expression of *mmpL3*, which encodes the TMM export system, and observed a significant downregulation during the entire course of PLB formation (Supplementary Fig. 5a). This implied that MmpL3-mediated TMM exporting activity was functionally dispensable during PLB formation.

**The trehalose-catalytic shift maintains ATP and NADPH of PLB**. The metabolomics profile of ΔtreS PLB demonstrated an inability to biosynthesize maltose, an initial metabolite for the trehalose-catalytic shift (Fig. 2a). We speculated that the inability of ΔtreS to form intact PLB was largely attributed to its failure to maintain the downstream pathways fueled by the trehalose-catalytic shift and their functional deficiencies. Supplementation

with exogenous 10–25 mM maltose was sufficient to restore ΔtreS PLB formation up to ~90% of WT PLB (Fig. 3a). Maltose-mediated rescue of ΔtreS PLB formation was also accompanied by a dose-dependent restoration of glucose-P and GL/PPP intermediates (Fig. 3a, Supplementary Fig. 6a), confirming that GL and the PPP are two primary downstream pathways. Using a mixture of 90% ¹²C and 10% universally ¹³C labeled ([U-¹³C]) maltose, we observed specific ¹³C enrichment of fully labeled (M + 6), with minor M + 3, glucose-P isotopolog in ΔtreS PLB, revealing that maltose serves as a direct or indirect substrate of the trehalose-catalytic shift (Supplementary Fig. 6b). As GL and the PPP are known to be metabolic sources for ATP and NADPH, respectively, we sought to prove that trehalose-mediated carbon flux toward GL and the PPP is a compensatory mechanism to maintain levels of ATP and NADPH. Unlike WT or ΔtreS/com PLB, the ATP level of ΔtreS was ~70% of the ATP level in the replicating state (day 0 of the biofilm), rapidly decreased during the initiation stage (days 0–16), and was undetectable at day 22 with slightly recovering thereafter (days 28–33; Fig. 3c), which was fully restored by 25 mM maltose single supplementation at the initiation stage (Fig. 3d). This indicates that the trehalose-catalytic shift helps maintain ATP levels even while lacking full ETC-mediated ATP biosynthetic activity. Similar to ATP kinetics, NADPH levels in ΔtreS at day 28 were significantly lower than those of WT Mtb, but were completely restored by supplementing with 25 mM maltose (Supplementary Fig. 6c). Since NADPH is an electron donor for biosynthesis of γ-glutamylcysteine, a sulfur donor for the antioxidant, ergothioneine[38,39], γ-glutamylcysteine abundance was significantly impaired in ΔtreS PLB, while restoring by adding 25 mM maltose (Supplementary Fig. 6d). This suggests that ΔtreS lacks adaptive strategies used to defend ROS-mediated damage. To examine the role of PPP-mediated antioxidant biosynthesis in PLB formation, we generated an Mtb mutant lacking the second enzyme of the PPP, 6-phosphogluconolactonase (ΔdevB; Supplementary Fig. 6e, f) and monitored ΔdevB PLB formation using biofilm culture. Surprisingly, ΔdevB was able to form PLB at levels similar to WT PLB (Supplementary Fig. 6g). This suggested that the primary role of the trehalose-catalytic shift is to secure alternate carbon sources to compensate primarily for ATP levels, but also for antioxidant depletion.

**The trehalose-catalytic shift mitigates antibiotic effect**. BDQ, a new FDA-approved TB antibiotic, inhibits energy metabolism by targeting ATP biosynthesis. Since ATP depletion together with perturbations in the ETC was shown to trigger bacterial persister formation[20,22], we hypothesized that weak early bactericidal effects of BDQ on Mtb are attributed to a TreS-mediated trehalose-catalytic shift. To examine this, we treated WT Mtb, ΔtreS, and ΔtreS/com (complemented strain) with 10× MIC equivalent of BDQ and CCCP (a membrane potential dissipating agent). TreS deficiency resulted in an ~1.5 log₁₀ decrease in CFU within 2 days and ultimately ~144-fold and ~111-fold greater sensitivity to BDQ and CCCP, respectively, at day 8 of incubation (Fig. 4a, Supplementary Fig. 7a). This enhanced sensitivity of ΔtreS to BDQ was associated with a rapid depletion in ATP compared with that of WT or ΔtreS/com, likely due to a lack of a trehalose-catalytic shift, and an accompanied defect in ATP compensation (Supplementary Fig. 7b). Metabolomics profiles of ΔtreS following BDQ treatment showed accumulation of TMM/TDM, no discernible biosynthesis of maltose, and a defect in glucose-P and sedoheptulose phosphate (S7P) production (Supplementary Fig. 8a, b). As seen in PLB, WT following BDQ treatment also resulted in specific induction of *treS* mRNA by ~8 fold, with minor changes in *otsA*, *otsB1*, or *otsB2* (Supplementary Fig. 7c).

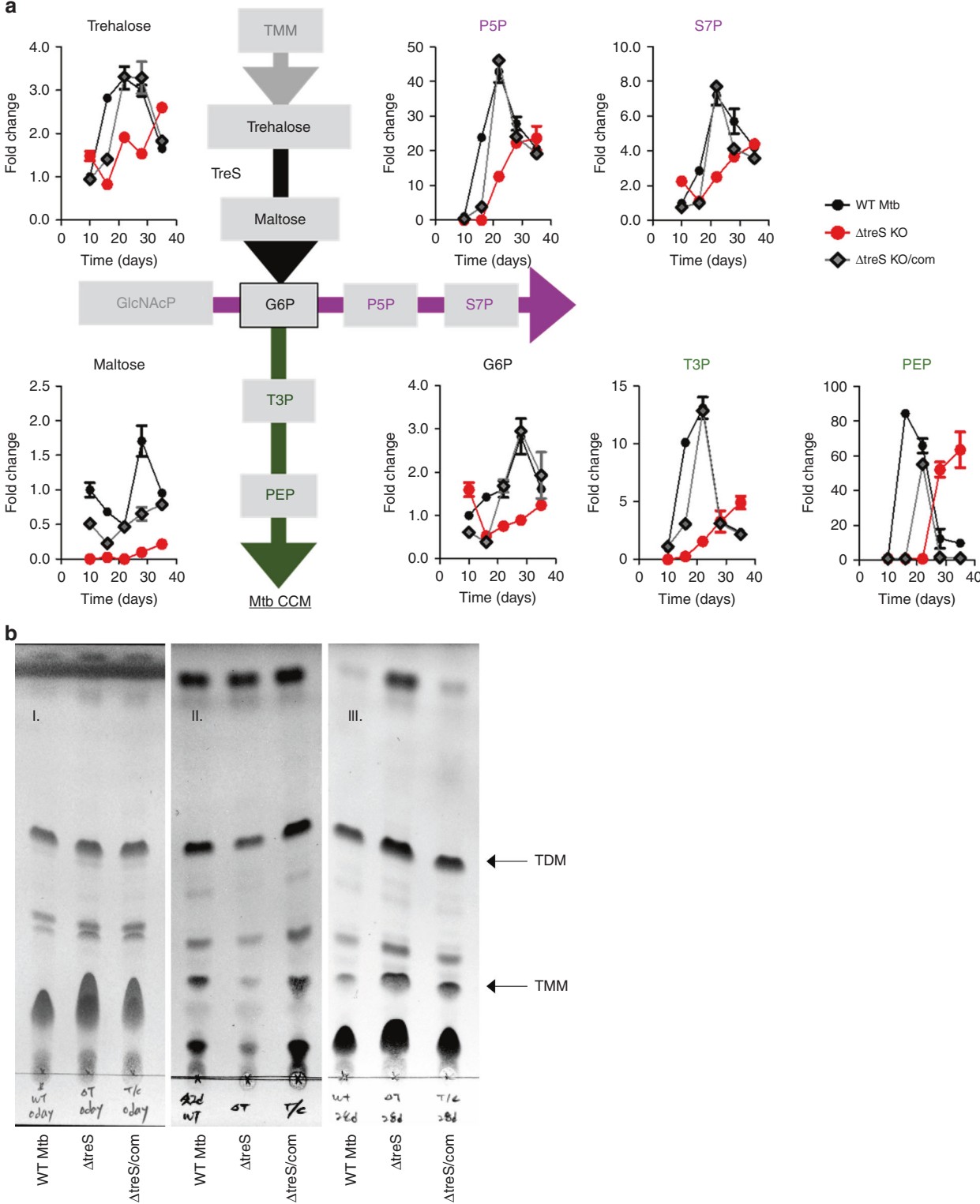

**Fig. 2** Metabolic consequences of TreS deficiency on PLB metabolism. **a** Time-course kinetics of pool sizes of glycolysis (green font/line), the pentose phosphate pathway (magenta font/line), and sucrose metabolism (black font/line) intermediates of WT Mtb, ΔtreS, and ΔtreS/com of mycobacterial biofilm culture. P5P, pentose 5-phosphates; S7P, sedoheptulose 7-phosphates; G6P, glucose 6-phosphates; T3P, triose 3-phosphates (dihydroxyacetone phosphate and glyceraldehyde phosphate); PEP, phosphoenolpyruvate. **b** Time course of extractable TMM/TDM from WT Mtb, ΔtreS, and ΔtreS/com harvested during the replicating state (**I**), day 22 (**II**), and day 28 (**III**) of mycobacterial biofilm culture. TMM/TDM extraction and TLC development and visualization were conducted as described in Fig. 1c. All values are the average of two independent biological duplicates ±s.e.m. Source data are provided as a Source Data file

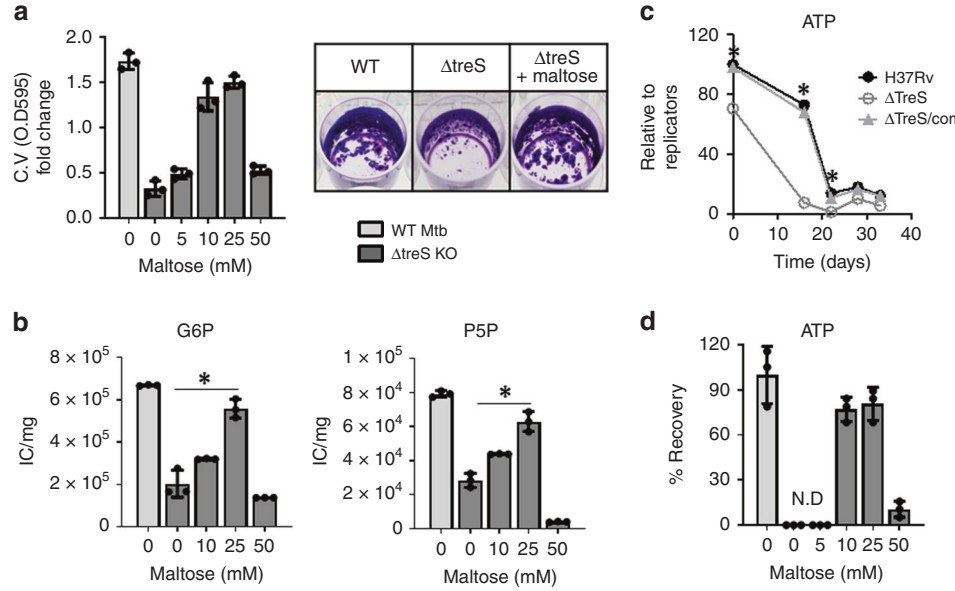

**Fig. 3** Maltose-chemical rescue of ΔtreS PLB formation. **a** CV staining OD$_{595}$ values of 28-day-old ΔtreS PLB fraction generated within mycobacterial biofilm culture in the presence of 0–50 mM maltose (left panel), and CV staining patterns of WT Mtb, ΔtreS, and ΔtreS treated with 20 mM maltose (right panel). **b** The effects of supplementing various concentrations of maltose on the ΔtreS PLB metabolome. Total bar heights indicate intrabacterial pool sizes compared with ΔtreS with no maltose supplement or WT Mtb. **c** Time course of intrabacterial ATP levels of WT Mtb (H$_{37}$Rv), ΔtreS, and ΔtreS/com while incubating within mycobacterial biofilm culture. **d** The effect of supplementing ΔtreS PLB fraction with 20 mM maltose on intrabacterial ATP levels at day 28. All values are the average of biological triplicates ± s.e.m. *$P < 0.001$ by Student's unpaired *t*-test. G6P, glucose 6-phosphates; P5P, pentose 5-phosphates. Source data are provided as a Source Data file

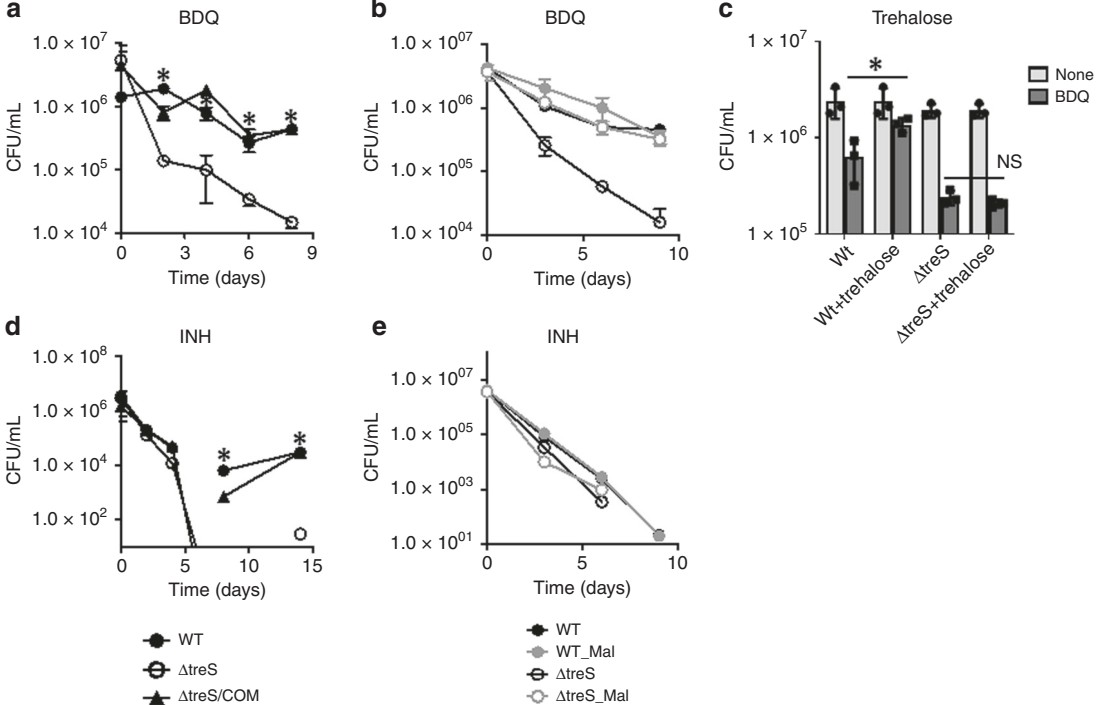

**Fig. 4** The trehalose-catalytic shift is used to circumvent BDQ and INH effects. **a** CFU (colony-forming unit) viability of WT Mtb, ΔtreS, and ΔtreS/com following treatment with 10× MIC-equivalent BDQ. The effect of supplementing with **b** 20 mM maltose and **c** trehalose on WT Mtb ΔtreS, and ΔtreS/com CFU viability following treatment with 10× MIC-equivalent BDQ. **d** CFU viability of WT Mtb, ΔtreS, and ΔtreS/com following treatment with 10× MIC-equivalent INH. **e** The effect of supplementation of 20 mM maltose on CFU viability of WT Mtb, ΔtreS, and ΔtreS/com with co-treatment of 10× MIC-equivalent INH. All values are the average of experimental triplicates ±s.e.m. and representative of at least two independent experiments. *$P < 0.01$. NS, not significant by Student's unpaired *t*-test

Furthermore, supplementation with 20 mM maltose, but not trehalose, was sufficient to render ΔtreS as tolerant as WT or ΔtreS/com to BDQ or CCCP (Fig. 4b, c). The metabolomics profile of ΔtreS PLB demonstrated that TreS deficiency increased ROS damage (Supplementary Fig. 8a, b). Thus, we examined the effect of co-incubation of 10× MIC-equivalent BDQ with the thiol-based antioxidant thiourea on ΔtreS. Thiourea afforded a roughly 1.5-$\log_{10}$ increase in CFU, approaching near-complete protection (Supplementary Fig. 7d). A greater sensitivity of ΔtreS as compared with WT Mtb was also observed when 5 mM $H_2O_2$ was used as an ROS generator (Supplementary Fig. 7e). Collectively, Mtb uses the trehalose-catalytic shift to resist BDQ- or CCCP-mediated antibiotic effects.

Unlike the differences in sensitivity against BDQ or CCCP, all three strains showed identical susceptibility against 10× MIC-equivalent INH or RIF and almost complete clearance at day 6 of incubation (Fig. 4d, Supplementary Fig. 9a). Exogenous supplementation with maltose had little effect on ΔtreS-drug sensitivity to INH (Fig. 4e). Intriguingly, at 8 or 14 days post treatment with INH or RIF, respectively, WT colonies reappeared, while ΔtreS failed to form viable colonies (Fig. 4d, Supplementary Fig. 9a), implying that the trehalose-catalytic shift is required for the formation of viable but nonculturable cells (VBNC), another Mtb persister-like state that arises following antibiotic treatment[6,40]. To confirm the requirement of the trehalose-catalytic shift in VBNC formation, we treated Mtb with a high dose of INH or RIF and monitored outgrowth[41]. A 10-day incubation of 30× MIC-equivalent INH or RIF prevented colony formation in all three strains, but a 21-fold diluted culture of WT in fresh antibiotic-free m7H9 media resulted in regrowth. However, ΔtreS, or WT treated with the specific TreS inhibitor, validamycin A (Val A)[42], showed a significant delay in outgrowth (Supplementary Fig. 9b). Taken together, the trehalose-catalytic shift not only contributes to circumventing the energy metabolism inhibition caused by BDQ or CCCP, but also to forming VBNC following first-line TB drug treatment.

**6-treAz inhibits TreS-mediated trehalose-catalytic shift**. A series of trehalose analogs, including 6-treAz (6-azido-6-α,α'-trehalose), were recently synthesized chemoenzymatically and shown to have selective inhibitory activity against *M. smegmatis* biofilm formation with no discernible interference in planktonic growth (Supplementary Fig. 10a)[43,44]. Based upon the structural similarity between 6-treAz and trehalose, we hypothesized that the anti-mycobacterial biofilm activity of 6-treAz is attributable to its competitive inhibition of the TreS-mediated trehalose-catalytic shift. Indeed, treatment of the in vitro TreS enzyme reaction with 100 μM 6-treAz for up to 4 h efficiently abolished maltose production (Supplementary Fig. 10b). Consequently, treatment with at least 100 μM 6-treAz to WT Mtb biofilm culture significantly interfered with PLB formation (Fig. 5a, Supplementary Fig. 10c). Overall, 6-treAz-mediated inhibition of PLB formation was accompanied by a dose-dependent inhibition of the trehalose-catalytic shift, in which we observed gradual accumulation of trehalose with reciprocal depletion of maltose, glucose-P, and S7P (Fig. 5a, b). To claim 6-treAz as a potential candidate for adjunctive treatment, we treated WT Mtb with 100 μM 6-treAz and 10× MIC-equivalent BDQ for 9 days and observed that WT Mtb displayed rapid and heightened sensitivity to BDQ by ~2-$\log_{10}$ CFU, a level similar to that of ΔtreS (Fig. 5c). No further effects of 6-treAz on ΔtreS susceptibility to BDQ were observed, confirming the specificity of 6-treAz as a TreS inhibitor (Fig. 5c). Using macrophages derived from human peripheral blood mononuclear cells (PBMC), we also observed that co-treatment of 100 μM 6-treAz and 10× MIC-equivalent BDQ boosted the sensitivity to BDQ by ~1.5-$\log_{10}$ CFU within 3 days (Fig. 5d). Thus, TreS and the trehalose-catalytic shift can serve as potential sources to develop adjunctive therapeutics that

will enhance the effects of the conventional TB drug regimen. Trehalose analogs can be initial structures for further development of TreS-specific inhibitors.

**Drug-resistant Mtb executes the trehalose-catalytic shift**. To determine if the trehalose-catalytic shift plays a role in clinical TB isolates' drug resistance, we collected three drug-sensitive TB isolates (KT0294, KT383, and KT0385), two XDR-TB isolates (KT0134 and KT0384), and one TDR-TB isolate (KT1111) (Supplementary Table 2) from the clinical TB isolate library available in the ITRC (International Tuberculosis Research Center). The growth of TDR-TB isolates was significantly impaired when media contained either glycerol or glucose as a single carbon source, while drug-sensitive TB isolates grew at levels relatively similar to those of $H_{37}Rv$ (Supplementary Fig. 11a). Only 10 mM sodium butyrate (SB) supported the growth of all clinical TB isolates at levels comparable with $H_{37}Rv$ (Fig. 6a, b, Supplementary Fig. 11a–c). The specificity of growth-permissive carbon sources implied that XDR- and TDR-TB isolates underwent significant alterations in their CCM activities, due to antibiotic and/or immunological pressures, while also accumulating drug-resistant mutations. To assess whether altered CCM activities were due to the trehalose-catalytic shift, we monitored in vitro growth patterns of all TB isolates following supplementation with 10 mM trehalose to media that already contained 10 mM SB. SB-mediated growth of all TB isolates was further enhanced by trehalose, indicating that exogenous trehalose can be used by both drug-sensitive and drug-resistant TB isolates (Fig. 6a, b, Supplementary Fig. 11b, c). Intriguingly, trehalose-mediated growth enhancement of XDR- and TDR-TB isolates was suppressed by treatment with 100 μM Val A, a TreS inhibitor, to the same exact level as that of SB only, while the growth enhancement of drug-sensitive TB isolates was relatively unaffected by Val A treatment (Fig. 6a, b). The specific sensitivity of XDR- and TDR-TB isolates to Val A-mediated TreS inhibition indicated that XDR- and TDR-TB isolates underwent TreS-centered CCM remodeling, which did not yet occur in drug-sensitive TB isolates. These TreS-centered CCM alterations in XDR- and TDR-TB isolates were also confirmed by metabolomics profiles (Fig. 7) and TLC-based lipid profiles (Supplementary Fig. 11d). Supplementing increasing doses of trehalose resulted in a gradual accumulation of trehalose. However, the levels were significantly greater in drug-sensitive TB isolates as compared with those of XDR- or TDR-TB isolates, presumably due to a reduction in trehalose uptake by XDR- or TDR-TB isolates. Subsequently, maltose, glucose-P, S7P, and PEP were induced in all TB isolates, but their induction levels were different between the drug-sensitive and drug-resistant (XDR- and TDR-) groups (Fig. 7). To show the evidence of trehalose-catalytic shift, we calculated the conversion rates to maltose, glucose-P, S7P, and PEP from newly intake trehalose. Indeed, XDR- and TDR-TB isolates showed greater conversion rates than those of drug-sensitive TB isolates (Fig. 7). More substantial trehalose-catalytic shift in XDR- and TDR-TB isolates was also confirmed by the maintenance or reduction of TMM/TDM in response to trehalose supplementation, especially in KT0384 and KT1111, while drug-sensitive TB isolates showed an induction of TMM/TDM (Supplementary Fig. 11d). Intriguingly, trehalose supplementation-associated TMM/TDM changes were the weakest in KT0134, which kinetically matched the lowest trehalose-to-maltose conversion rate among three drug-resistant TB isolates (Fig. 7 and Supplementary Fig. 11d). Foregoing growth patterns, metabolomics and lipidomics of drug-sensitive and drug-resistant TB clinical isolates indicated that the TreS-mediated trehalose-catalytic shift is active in XDR- and TDR-TB isolates. These findings

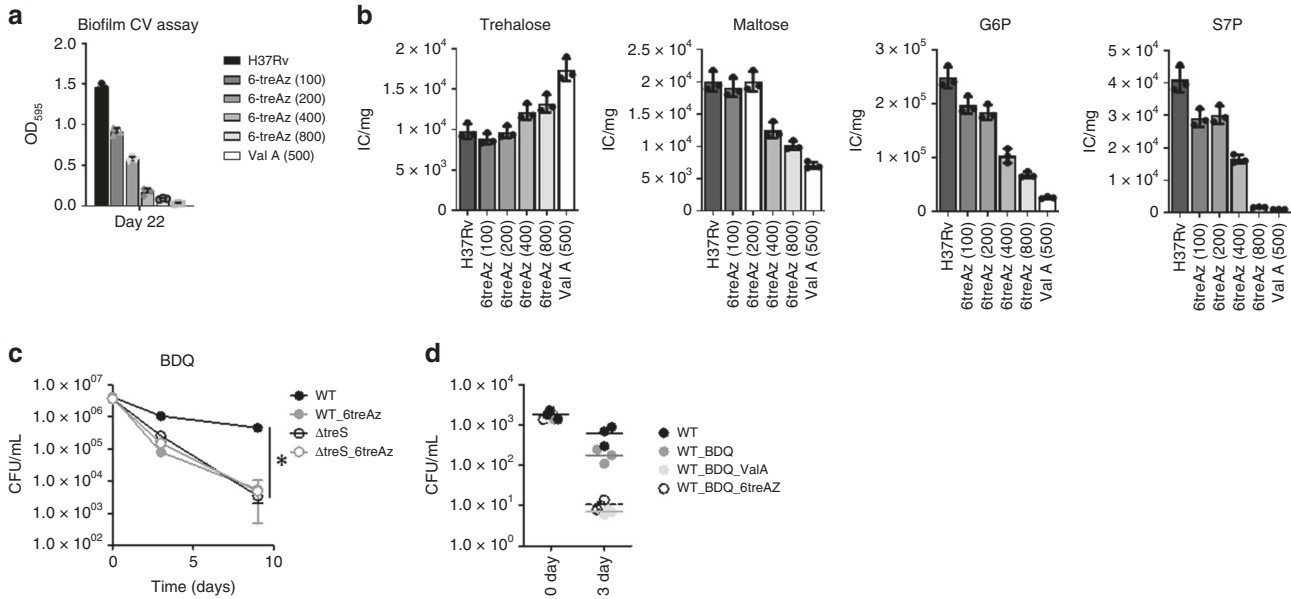

**Fig. 5** Anti-PLB formation activity of 6-azido-6-α,α'-trehalose (6-treAz). **a** The effects of various concentrations (100–800 μM) of 6-treAz treatment on PLB formation were monitored by CV staining assay of PLB harvested at day 28 of mycobacterial biofilm culture. Treatment with 500 μM validamycin A was included as a positive control. **b** Metabolic impact of various concentrations of 6-treAz on PLB metabolism. The metabolome of PLB was harvested at day 28 PLB within mycobacterial biofilm culture. Intrabacterial pool sizes of PLB intermediates in trehalose metabolism (trehalose and maltose) and GL (G6P) and PPP (S7P) following treatment with various concentrations of 6-treAz or 500 μM Val A. **c** CFU-based viability of WT Mtb or ΔtreS in the presence or absence of 200 μM 6-treAz following co-treatment with 10× MIC equivalent of BDQ. **d** The effect of 6-treAz and/or Val A on CFU-based viability of WT Mtb inside human PBMC macrophages following treatment with 10× MIC-equivalent BDQ. All values are the average of biological triplicates ±s.e.m. *$P < 0.001$ by ANOVA with Bonferroni post-test correction. S7P, sedoheptulose 7-phosphates; G6P, glucose 6-phosphates. Source data are provided as a Source Data file

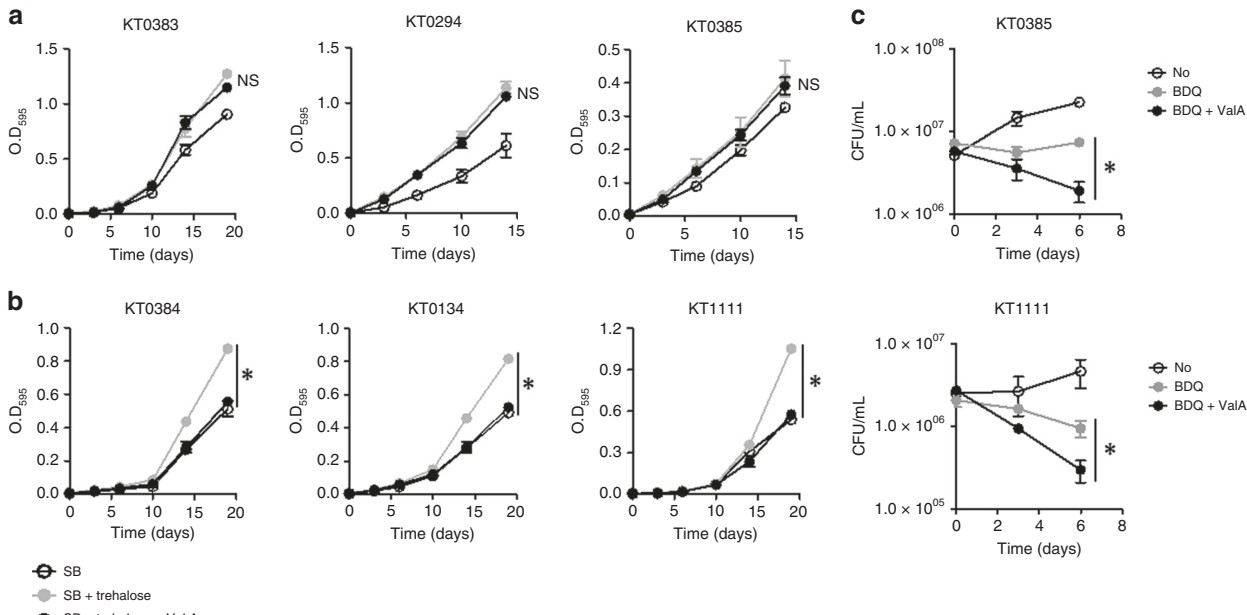

**Fig. 6** XDR- and TDR-TB clinical isolates use the trehalose-catalytic shift for survival. **a** The growth of drug-sensitive TB isolates (KT0294, KT0383, and KT0385) in m7H9 media containing 10 mM sodium butyrate (SB) was enhanced by supplementing with 20 mM trehalose, but not restored by treating with 200 μM validamycin A (Val A). **b** The growth of drug-resistant isolates (XDR-TB, KT0134, and KT0384; TDR-TB, KT1111) in m7H9 media containing 10 mM SB was enhanced by supplementing with 20 mM trehalose, and completely restored to trehalose-untreated levels following treatment with 200 μM Val A. **c** Chemical inhibition of TreS as an adjunctive therapeutic strategy to boost BDQ-mediated antimicrobial effect on drug-sensitive and drug-resistant clinical isolates. All values are the average of biological triplicates ±s.e.m. *$P < 0.01$; NS, not significant by ANOVA between SB + trehalose and SB + trehalose + Val A (**a**) and between BDQ and BDQ + Val A (**b**) with Bonferroni post-test correction

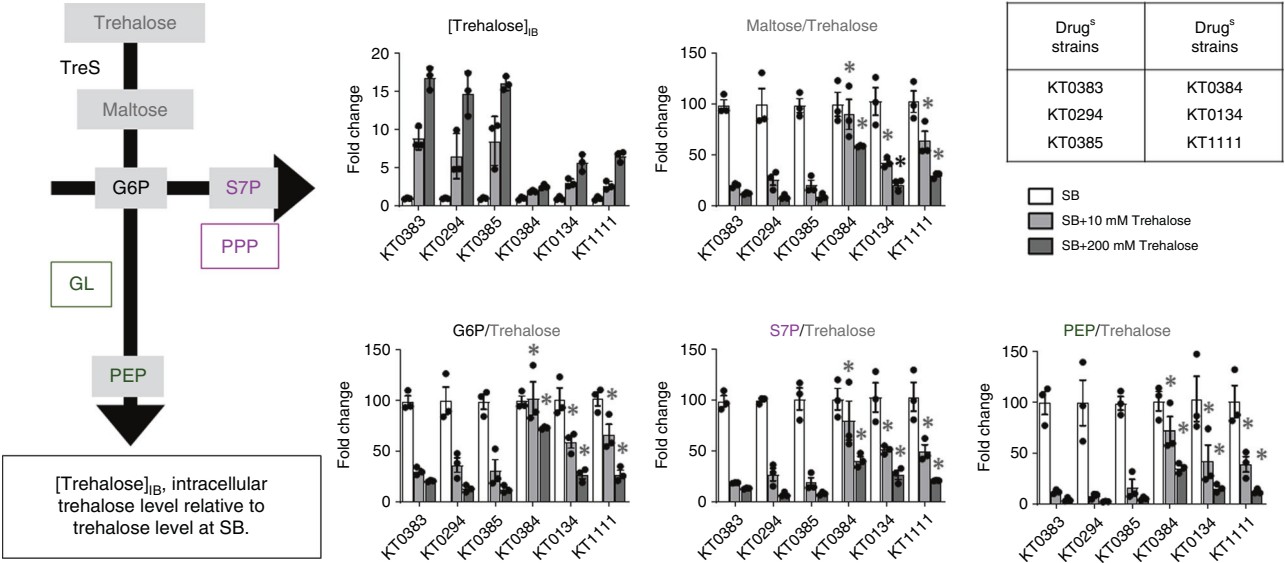

**Fig. 7** The effect of supplementing trehalose on TB clinical isolate metabolism. Supplementation with 10 or 20 mM trehalose on the central metabolic pathways of drug-sensitive, Drug$^S$, (KT0294, KT0383, and KT0385), and drug-resistant, Drug$^R$, XDR- (KT0134 and KT0384), and TDR- (KT1111) TB clinical isolates was monitored by metabolomics. Left panel, trehalose metabolism, glycolysis (GL), and the pentose phosphate pathway (PPP) intermediates were analyzed. Right panel, [Trehalose]$_{IB}$ showed intrabacterial trehalose detected within each TB clinical isolate strain following exposure to 0, 10, or 20 mM trehalose supplementation for 24 h. Metabolite fold change describes the percent conversion rate using intake trehalose under each condition, including 0, 10, and 20 mM trehalose supplementation. The conversion rate of each metabolite from trehalose under SB (sodium butyrate with 0 mM trehalose) was set to 100%. All values are the average of biological triplicates ±s.e.m. *$P < 0.01$ by Student's unpaired t-test (between drug-resistant TB isolate metabolites and the average of the corresponding metabolite of drug-sensitive TB isolates; KT0294, KT0384, and KT0385). Data shown represent the combined results of duplicates of three replicates. Source data are provided as a Source Data file

suggest that the trehalose-catalytic shift may be both transient and permanent survival strategies utilized by Mtb.

Consistent with the foregoing metabolomics finding that trehalose-catalytic shift has undergone in clinical TB isolates (Fig. 7 and Supplementary Fig. 11), 20 mM trehalose efficiently protected BDQ sensitivities of all TB isolates and treatment with Val A boosted the BDQ-bactericidal effects on all TB isolates. We also observed that Val A afforded more rapid and vigorous depletion in CFUs of clinical TB isolates, as compared with those of BDQ single treatment (Fig. 6c). Thus, the trehalose-catalytic shift can serve as a source of adjunctive options used in combination with the conventional TB drug regimen to treat both drug-sensitive and drug-resistant TB infections.

## Discussion

Bacterial pathogens including Mtb employ diverse adaptive strategies to survive antibiotic-treatment conditions[45–47]. In this study, we demonstrated that Mtb altered its trehalose metabolism as an adaptive metabolic strategy in order to achieve transient drug tolerance. The stresses associated with Mtb PLB formation within our biofilm system were associated with systemic metabolic damage due to nutrient starvation, intrabacterial ATP depletion, ETC dysregulation, and accompanied loss of redox homeostasis[48]. In order to overcome these challenges, preexisting trehalose was internally catalyzed via altered metabolic activities, such that its carbon flux toward the biosynthesis of GL and PPP intermediates increased, while the carbon flux for the biosynthesis of cell surface TMM/TDM decreased. Indeed, both *mmpL3* mRNA expression and TMM abundance were downregulated (Fig. 2b, Supplementary Fig. 5a), suggesting that MmpL3 activity required for TMM export across the inner membrane was significantly downregulated. This was corroborated by previous findings demonstrating that MmpL3 inhibitors have an antimicrobial effect only on replicating Mtb, and not on hypoxic or non-replicating Mtb[49,50]. The trehalose-catalytic

shift produced initial substrates for GL and PPP intermediates and served as alternate biosynthetic sources of ATP, NADPH, and antioxidants. A recent multiomics study showed that BDQ-mediated ETC downregulation and ATP depletion induced Mtb dormancy regulon and BDQ-associated bactericidal effects were significantly impeded when BDQ was co-treated with glycolytic, but not non-fermentable, carbon sources. These implied that ATP depletion serves as a signal associated with survival as Mtb persisters, and metabolic adaptation to GL-mediated carbon flux helps mitigate the effects of BDQ[51]. Recently, Collins et al. found that an epidemic *Clostridium difficile* strain altered its trehalose metabolism to enhance its drug tolerance[52]. Here, we revealed that Mtb PLB executes the trehalose-catalytic shift in a manner that is functionally analogous to trehalose metabolism of virulent *C. difficile* and uncovers mechanistic bases underlying Mtb adaptive strategies to circumvent BDQ-treated damage. Intriguingly, the same metabolic activity was observed in XDR- and TDR-, but was weaker in drug-sensitive clinical TB isolates, implying a functional link between the trehalose-catalytic shift, acquisition of transient drug tolerance, and emergence of permanent drug resistance.

Whether permanent drug resistance evolves from transient drug tolerance, and how drug tolerance accelerates the evolution of drug resistance are the subject of debate. Recent genetic studies using *Escherichia coli* showed that intermittent exposure to ampicillin significantly delayed *E. coli* lag time prior to increased ampicillin MIC[53,54]. This suggested that *E. coli* became tolerant to several cycles ahead of drug resistance. Bactericidal antibiotics stimulate ROS production within the target bacteria[55–57] and a sublethal dose of ROS interferes with respiration and causes DNA damage, ultimately leading to accumulation of genetic mutations associated with drug resistance. Collectively, bactericidal antibiotics act as reactive mutagens, and repetitive cycling into and out of the persister state ultimately results in emergence of drug-resistant mutants. Our metabolomics profiles of PLB, and XDR-, and TDR-TB isolates supported this notion by showing that

drug-resistant TB isolates manage their CCM by executing the trehalose-catalytic shift, albeit the activity was weaker in drug-sensitive TB isolates. Thus, the trehalose-catalytic shift may be an attempt not only for transient survival in response to anti-microbials but also for permanent survival strategies through accumulation of genetic mutations.

Programmed evolution during antibiotic exposure helps explain the emergence of antibiotic-resistant mutations in bacterial pathogens[54]. Efforts to investigate the adaptive metabolic evolution required for drug-resistant mutants from a drug-sensitive TB strain have heavily relied on transcriptional profiling studies and the phenotypic consequences of genetic knockouts. This study uses metabolomics profiling and reveals that Mtb persisters use the catalytic shift of preexisting trehalose as an adaptive strategy to overcome immediate environmental challenges, form transient drug-tolerant persisters, and help accumulate permanent drug-resistant mutations (Supplementary Fig. 12). Thus, the trehalose-catalytic shift in Mtb persisters seems to be relatively unaffected by nutrient conditions within the surrounding environment. Collectively, this adaptation is somewhat analogous to the mammalian hibernation strategy, in which some animals, such as bears, endure environmentally challenging winters by stopping consumption of food but maintaining their essential metabolic activities, such as energy and antioxidant biosynthesis, through using preexisting brown fat[58].

From a clinical perspective, persister formation is undetectable with conventional tools, as persisters are genetically identical to the non-persister population. The results of this study highlight a pharmacologically valuable source to develop adjunctive treatment, with which we anticipate enhancing the antimicrobial outcomes of our conventional TB treatment by interfering with adaptive strategies. In addition, the trehalose-catalysis shift may also provide a source of biomarkers to discriminate patients infected with drug-resistant isolates and thereby determine future therapeutic directions and predict their outcomes.

## Methods

**Bacterial strains and culture conditions**. Mtb H$_{37}$Rv, *treS* knockout (ΔtreS) and the complemented strain (ΔtreS/com), *otsB1* knockout (ΔotsB1), *rv2402* knockout (Δrv2402), *devB* knockout (ΔdevB), three drug-sensitive TB isolates (KT0294, KT0383, and KT0385), two XDR-TB isolates (KT0134 and KT0384), and one TDR-TB isolate (KT1111) were cultured in a biosafety level 3 facility at 37 °C in Middlebrook 7H9 broth (m7H9) (Difco), in Sauton minimal media, or on Middlebrook 7H10 agar (m7H10) (Difco) supplemented with 0.04% Tyloxapol (planktonic growth in broth only), 0.5 g L$^{-1}$ BSA (Fraction V), and 0.085% NaCl. When appropriate, hygromycin B (50 μg mL$^{-1}$) and kanamycin (30 μg mL$^{-1}$) were used. Drug-sensitive and two XDR-TB clinical isolates were originally isolated from the sputum of TB patients, and TDR-TB clinical isolate was collected from the dissected lung cavity of incurable TB patients who underwent lung resection surgery at the National Masan Tuberculosis Hospital (NMTH). We have complied with all relevant ethical regulations for work with TB clinical isolates. Human tissue collection was approved by Institutional Review Board of NMTH (project license: IRB-05-I-N069 and IRB-05-E01) and written informed consent was obtained from the patients for collection and use of the specimens for research. For survival assays, bacterial cultures were taken from growth curve cultures at the time points indicated and plated for CFU-based enumeration assays.

**Mtb filter culture and metabolite extraction**. Mtb-laden filters used for metabolomics profiling were generated[47] and incubated at 37 °C for 5 days to reach the mid-log phase of growth. MIC values of TB antibiotics were determined using a m7H9. In total, 10× MIC bioequivalent doses of antibiotics were used to expose Mtb grown atop agar-supported filters at larger inocula suitable for filter-culture-based metabolomics profiling platform. Mtb-laden filters were metabolically quenched by plunging filters into a mixture of acetonitrile:methanol:H$_2$O (40:40:20, v:v:v) precooled to –40 °C, and metabolites were extracted by mechanical lysis with 0.1-mm zirconia beads in Precellys tissue homogenizer for 4 min (6000 rpm) twice under continuous cooling at or below 2 °C. Lysates were clarified by centrifugation and then filtered across a 0.22-μm spin-X column. The residual protein content of metabolite extracts was determined (BCA protein assay kit, Thermo Scientific) to normalize the samples to cell biomass.

**devB-deficient Mtb construction**. The *devB*-deficient knockout mutant (ΔdevB) was generated in H$_{37}$Rv background[28]. ΔdevB was constructed via a suicide plasmid approach (Fig. S6f). Approximately 500-bp fragments corresponding to regions upstream and downstream of *devB* gene were amplified by PCR and cloned into a temperature-sensitive vector pDE43-XSTS to flank the hygromycin-resistance cassette and generate pKO-XSTS-*devB*. H$_{37}$Rv was first transformed with pKO-XSTS-*devB* and plated on m7H10 with hygromycin at the permissive temperature of 30 °C. Transformants were then grown at 30 °C to OD$_{595}$ = ~1.0 and plated on m7H10 with hygromycin, 10% sucrose, 0.2% glycerol, and 0.2% glucose at the restrictive temperature of 40 °C. For complementation of ΔdevB, we generated pGMEK-phsp60-*devB*, a vector that expresses *devB* under the control of the hsp60 promotor and WT DevB is constitutively expressed episomally. The removal of native *devB* in H$_{37}$Rv was confirmed by Southern blot analysis. Uncropped Southern blot film picture is available in the Source Data.

**Drug-susceptibility test for clinical TB isolates**. Drug-sensitive or resistant TB isolates were processed for phenotypic drug-susceptibility test (DST) on Löwenstein–Jensen (LJ) plates containing a panel of TB drugs (Supplementary Table 2) at absolute concentrations. The critical concentrations of each drug for determining resistance were 0.2 μg mL$^{-1}$ of INH, 40 μg mL$^{-1}$ RIF, 2.0 μg mL$^{-1}$ of ethambutol (EMB), 10 μg mL$^{-1}$ of streptomycin (SM), 40 μg mL$^{-1}$ of kanamycin (KM), 40 μg mL$^{-1}$ of prothionamide (PTH), 30 μg mL$^{-1}$ of DCS, 1.0 μg mL$^{-1}$ of p-aminosalicylic acid (PAS), 2.0 μg mL$^{-1}$ of ofloxacin (OFX), and 50 μg mL$^{-1}$ of pyrazinamide (PZA). In the case of PZA, the LJ medium was readjusted to pH 5.7 before usage, and a control medium plate without PZA was also used as a control. Drug resistance was verified when more than 1% of colonies of TB isolates grew on drug-containing LJ medium.

In addition to LJ medium-mediated DST, drug susceptibility was separately confirmed using BACTEC MGIT (BD Microbiology Systems). For the DST using MGIT, the drugs were used following the standard procedures of the manufacturers. The final drug concentrations were 0.1 μg mL$^{-1}$ of INH, 1.0 μg mL$^{-1}$ of RIF, 5.0 μg mL$^{-1}$ of EMB, 1.0 μg mL$^{-1}$ of SM, 2.0 μg mL$^{-1}$ of OFX, 0.25 μg mL$^{-1}$ of moxifloxacin (MFX), 2.5 μg mL$^{-1}$ of KM, and 2.5 μg mL$^{-1}$ of ciprofloxacin (CPM). For each clinical TB isolate, a growth control tube with m7H9 without drug was included. The relative growth ratio between sample tubes and the corresponding control tubes was determined by the system software algorithm. The final interpretation and the susceptibility results were reported automatically.

**In vitro biofilm culture, CV assay, and metabolite extraction**. Mtb culture at mid-log phase in m7H9 was readjusted to an OD$_{595}$ of 1.0 and then diluted 1:100 (v:v) in fresh Sauton media without tyloxapol to get Mtb culture at OD$_{595}$ 0.01. Two milliliters of the culture was dispensed in each well in 24-well polystyrene plates (Eppendorf). The plates were wrapped with parafilm three times to limit gas exchange and incubated at 37 °C for around 5 weeks without agitation. At each time point, the extent of Mtb PLB formation was monitored by crystal violet staining and OD$_{595}$ measurements. The parafilm was gently unwrapped, and the media was removed underneath the biofilm (Supplementary Fig. 1a) using needle-syringe assembly. Wells were washed three times with PBS and dried. Dried PLB from biofilm culture was treated with 1 mL of 1% crystal violet (Sigma-Aldrich) for 10 min. Excess crystal violet was removed by washing three times with 1× PBS and then wells were dried again. In total, 1 mL of 95% ethanol was added to each well for 10 min. Then, OD$_{595}$ of 20-fold serial dilutions was read using a spectrophotometer.

At each time point after media removal, PLB of biofilm culture was washed with ice-cold PBS three times and quenched by adding 1 mL of acetonitrile:methanol: H$_2$O (40:40:20) precooled to –40 °C. Due to the small PLB biomass, metabolomics profiling was done in two independent biological duplicates and individual averages due to extremely low biomass. PLB metabolites were extracted by mechanical lysis with 0.1-mm zirconia beads in Precellys tissue homogenizer for 4 min (6000 rpm) twice under continuous cooling at or below 2 °C. Lysates were clarified by centrifugation and then filtered across a 0.22-μm spin-X column. The residual protein content of metabolite extracts was determined (BCA protein assay kit, Thermo Scientific) to normalize the samples to cell biomass.

**[U-$^{13}$C] maltose tracing**. The 24-well plates for biofilm culture using ΔtreS were initiated by supplementing with 20 mM maltose. At day 20, 20 mM of a mixture of 10% [U-$^{13}$C] maltose and 90% $^{12}$C maltose was added to the ΔtreS PLB fraction using needle-syringe assembly to prevent from perturbing PLB formation. The metabolites were sampled following incubation for an additional 2 days. The extent of isotopic labeling for glucose-P was determined by dividing the summed peak height ion intensities of all labeled isotopolog species by the ion intensity of both labeled and unlabeled species, expressed in percentage. Label-specific ion counts were corrected for naturally occurring $^{13}$C species (i.e., M + 1 and M + 2). The relative abundance of each isotopically labeled species was determined by dividing the peak height ion intensity of each isotopic form by the summed peak height ion intensity of all labeled species.

**LC-MS metabolomics profiling**. Extracted metabolites were separated on a Cogent Diamond Hydride Type C column (gradient 3) (Microsolve Technologies) and the mobile phase consisted of solution A (ddH$_2$O with 0.2% formic acid) and

solution B (acetonitrile with 0.2% formic acid). The mass spectrometer used was an Agilent Accurate Mass 6230 time of flight (TOF) coupled with an Agilent 1290 liquid chromatography (LC) system. Detected ions were deemed metabolites on the basis of unique accurate mass-retention time identifiers for masses exhibiting the expected distribution of accompanying isotopologs. The abundance of extracted metabolites was extracted using Agilent Qualitative Analysis B.07.00 and Profinder B.08.00 software (Agilent Technologies) with a mass tolerance of <0.005 Da. The clustered heatmap and hierarchical clustering trees were generated using Cluster 3.0 and Java Tree View 1.0. Principal component analysis was conducted using MetaboAnalyst (ver 4.0). All data, except for PLB, obtained by metabolomics profiling were the average of at least two independent triplicates.

**Lipid extraction and TLC analysis**. PLB was washed with cold PBS twice and the bacilli amount was measured by weight after drying. Then, PLB was transferred to a 15-mL amber glass bottle and contacted with 3 mL of chloroform:methanol (2:1, v:v) overnight to sterilize bacteria and extract total lipids simultaneously. A total of 10-mL acetone was added and further incubated for 24 h at –80 °C. After centrifugation, lipid extracts were decanted. Lipids were resuspended in 1 mL of chloroform:methanol (2:1, v:v) and equivalent amounts of each condition were loaded onto thin-layer chromatography (TLC) plates. TMM and TDM were resolved by TLC in the chlorofom:methanol:$H_2O$ (90:10:1, v:v:v) solvent system and visualized by spraying with 1% molybdophosphoric acid in ethanol and charring. Uncropped TLC figures are available in the Source Data.

**RNA extraction for qRT-PCR**. Mtb PLB formed within biofilm culture or following treatment with 100 $\mu$g mL$^{-1}$ d-cycloserine was harvested at various time points (days 16, 22, and 28 for biofilm culture and days 1, 7, and 14 for d-cycloserine treatment). The total RNA was extracted using TRIzol solution (Sigma-Aldrich) and mechanical lysing with 0.1-mm zirconia beads in a Precellys tissue homogenizer. Lysates were clarified by centrifugation and TRIzol supernatant was removed and used for RNA extraction. RNA was isolated using a Qiagen RNA extraction kit. Isolated RNA was treated with DNase I to remove DNA contamination (Sigma-Aldrich). RNA concentrations were determined using a Nanodrop, and qRT-PCR reactions were conducted using an iQ SYBR-Green Supermix (Bio-Rad) and C1000TM Thermal Cycler Instrument. Primers used for amplification are listed in Supplementary Table 1. Fold changes were calculated by $\Delta\Delta C_T$ values that were normalized to *sigA* transcript levels and depicted as log$_2$ values relative to planktonic growth with no antibiotic treatment.

**Measurement of intrabacterial ATP content and NADPH/NADP**. PLB of biofilm culture was resuspended in 1 mL of PBS containing 0.04% tyloxapol to generate single-cell suspension. Intrabacterial ATP concentrations were measured by BacTiter-Glo Microbial Cell Viability Assay (Promega) according to the manufacturer's instructions. NADP and NADPH concentrations were measured using a FluroNADP/NADPH detection kit (Cell Technology). Aliquots of 100 $\mu$L of bacterial cells were collected and metabolism of Mtb was rapidly quenched by plunging bacilli in the first solvent in each kit (cooled to <4 °C). Bioluminescence was determined in a plate reader. ATP bioluminescence was normalized by protein concentrations. ATP, NADP, and NADPH standards were also included in all of the experiments as internal control and standard curve generation.

**Measurement of PLB drug tolerance**. PLB at each time point of biofilm culture was resuspended in 1 mL of PBS containing 0.04% tyloxapol. Cultures were divided into two groups, one tube was "no-drug control" and the other tube was treated with 100× MIC equivalent with either INH (2 $\mu$g mL$^{-1}$) or RIF (3 $\mu$g mL$^{-1}$). Drug-treated cultures were incubated at 37 °C overnight and the relative cell viability was determined by comparing CFU relative to CFU of no-drug treatment control.

**CFU-based drug sensitivity of planktonic Mtb culture**. CFU-based cell enumeration assays were conducted in 96-well plates. Mid-log phase Mtb H$_{37}$Rv, $\Delta$treS, and $\Delta$treS/com cultures were diluted to an OD$_{595}$ of 0.05 (~1 × 10$^7$ cells per mL) in m7H9 in each well. TB drugs (INH, RIF, BDQ, or CCCP) and other supplements (maltose, trehalose, or thiourea) were added at the indicated concentrations and incubated for 1 or 2 weeks. Bacterial cultures were then serially diluted and plated on m7H10 agar, incubated for 3 weeks at 37 °C, and colonies were counted by hand.

**Outgrowth experiments**. Mid-log phase Mtb H$_{37}$Rv, $\Delta$treS, and $\Delta$treS/com cultures were resuspended in 5 mL of m7H9 with 30× MIC-equivalent INH or RIF and incubated at 37 °C for 10 days. The cultures were then diluted 21-fold (10 $\mu$L into 200 $\mu$L) into fresh m7H9 medium in a new 96-well plate (outgrowth plate). After 7 and 14 days, aliquots (100 $\mu$L) of the outgrowth plate culture will be transferred to clear-bottom plates for OD$_{595}$ determination. The impact on bacterial viability will be calculated by relative growth kinetics.

**Human monocyte isolation and differentiation into macrophages**. PBMCs were isolated from whole blood by using Sepmate according to the manufacturer's protocol. PBMCs were seeded in 10-cm cell culture dishes using IMDM medium supplemented with 15% FBS and 5% human AB serum. Monocytes were allowed to adhere in a 5% CO$_2$ incubator at 37 °C for 2 h. Non-adherent cells were removed, and the adherent cells were then washed, and resuspended in DMEM medium containing penicillin/streptomycin (1%), NEAA (1%), heat-inactivated FBS (10%), and sodium pyruvate (1%). Monocytes were seeded in 6-cm cell culture dishes at a density of 1 × 10$^6$ cells per mL and supplemented with GM-CSF (50 ng mL$^{-1}$) and cultivated for 7 additional days at 37 °C in a 5% CO$_2$ incubator to fully differentiate into monocyte-derived macrophages (MDM).

**MDM infection and CFU-enumeration assay**. Early mid-log phase Mtb culture was prepared using single-cell suspensions and dilution to a predefined viable number of bacilli in antibiotic-free RPMI. A total of 2 × 10$^5$ MDMs per well in 48-well plates were exposed to prepared single-cell suspensions of Mtb at MOI (multiplicity of infection) of 10. Four hours following exposure, uninfected bacilli were removed by washing with prewarmed PBS and treated with 30 $\mu$g mL$^{-1}$ gentamycin overnight. Infected MDMs were treated with 10× MIC-equivalent BDQ with or without 100 $\mu$M validamycin A or 100 $\mu$M 6-treAz. MDMs were lysed with 0.5% Triton X-100 and released bacilli were enumerated by plating serial dilutions of the lysates on m7H10 at 3-day post infection.

**In vitro TreS assay**. TreS activity was measured using a 100-$\mu$L in vitro enzyme reaction containing 40 mM MOPS (pH 7.0), 10 mM trehalose (Sigma-Aldrich), and 20 ng of purified TreS enzyme in the presence or absence of 100 $\mu$M 6-treAz or validamycin A (Santa Cruz Biotechnology). The reaction was initiated by adding TreS enzyme, incubated at 37 °C, and then quenched by heating and adding acetonitrile containing 0.2% formic acid to yield the final 70% acetonitrile mixture. After centrifugation, supernatants were used for LC–MS to detect maltose production.

**Statistical analyses**. Analyses were performed by an unpaired Student t-test and ANOVA test. P values <0.05 were considered statistically significant.

**Reporting summary**. Further information on research design is available in the Nature Research Reporting Summary linked to this article.

## Data availability
The data underlying Figs. 1a, 1c, 1d, 2a, 2b, 3a, 3b, 5b, and 7, Supplementary Figs. 1e, 2a, 3a, 5a, 5e, 6a, 6b, 6d, 6e, 7c, 8a, 8b, 10c, and 11d, and Supp Table 1 are provided as a Source Data file. All other data are available from the corresponding author upon reasonable request.

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

## Acknowledgements

This work was supported by start-up funding from the Department of Molecular Microbiology and Immunology, KSOM, USC, the Donald E. & Delia B. Baxter Foundation, the L.K. Whittier Foundation, Wright Foundation and NIH R21 AI139386 to H. E., NIH R15 AI117670 to B.M.S., and the Ministry of Health and Welfare of Republic of Korea to S.Y.E. and S.N.C.

## Author contributions

J.J.L., S.K.L., S.Y.E., and H.E. designed research. J.J.L., S.K.L., N.S., T.O.N., and B.M.S. performed experiments. J.J.L., S.K.L., S.E., S.N.C., and H.E. analyzed data and interpreted the results. J.J.L. and H.E. wrote the paper, which was edited by all authors. H.E. directed the research.

## Additional information

**Competing interests:** The authors declare no competing interests.

