## [Peer Review File · Nature Communications]

Reviewers' comments:

Reviewer #1 (Remarks to the Author):

Manuscript "Transient drug-tolerance and permanent drug-resistance rely 1 on the trehalose-catalytic shift in *Mycobacterium tuberculosis*"

This is an interesting, well-documented and comprehensive study on Mtb persisters generated during biofilm formation. The authors use Mtb grown in biofilm as a model for drug tolerance: the biofilm matrix is enriched in what the authors call "persister-like bacilli (PLB)". To validate PLB as Mtb persisters, the authors compared the mRNA profiles of PLB at different time with D-cycloserine (DCS)-treated Mtb culture, based on published work that d-cycloserine generates persisters in Mtb. Metabolomes of 200 metabolites were followed in the PLB formation over the course of 35 days. The authors found that trehalose metabolism and central carbon metabolism pathways were among the most altered pathways. The authors deleted *treS*, the gene involved in maltose conversion from trehalose and found that $\Delta treS$ failed to form PLB. Although the paper is interesting and describes a new way to look at Mtb persisters, numerous items need to be addressed before it is suitable for publication.

Major Comments:

- 1) The authors state that $\Delta treS$ was deficient in persister formation in planktonic conditions using DCS, yet Fig. S5d does not seem to reflect this statement. The killing of $\Delta treS$ with DCS follows the same biphasic killing than H37Rv suggesting that persisters are indeed formed in this condition. Although $\Delta treS$ is more sensitive to DCS, the strain may not be deficient in persister formation in planktonic condition.
- 2) The authors state that $\Delta treS$ may be more susceptible to ROS-mediated damage. This could be easily demonstrated by measuring CFUs/ml over time of $\Delta treS$ treated with hydrogen peroxide for example.
- 3) Lane218-221: The concept that thiourea acts as an antioxidant in Mtb is controversial. To prove their point, the authors should include a killing curve with a known ROS producer in combination with thiourea.
- 4) One of the authors conclusions is that the trehalose-catalytic shift serves as alternate biosynthetic sources of ATP, NADPH and antioxidants. A schematic of this alternate pathway should be added for clarification.

Minor comments

- 1) Add unit to figure S6c.

2) Either add the concentrations of BDQ and CCCP used in Fig. S7 or indicate what the MIC values are.

Reviewer #2 (Remarks to the Author):

The impressive study by Lee et al. shed lights the role of trehalose based lipids named TMM and TDM found in the cell envelope of Mycobacterium tuberculosis in the process of persisters bacilli and antimicrobial tolerance/resistance. This study is of high interest for the Tb community and can open new area of research of intracellular bacteria persisters.

The manuscript is clear, very-well written and the experiments have been conducted diligently with appropriate controls and validation.

The authors used an impressive armoury of state-of-the-art methods to address the hypothesis.

However, at this stage, it seems required that the authors address those points:

1) It is know that lipid bodies and triacylglycerol are also involved in antimicrobial tolerance. Based on the actual manuscript, have the authors investigated the presence of lipid bodies in the PLBs in addition to metabolome remodelling towards trehalose synthesis?

2) To be consistent with the proposed hypothesis, as TMM and TDM required also fatty acid synthesis I and II, have the authors also consider investigating the turn-over of acetyl-CoA and Krebs cycle intermediates in their metabolomics data? This can potential bring interesting results regarding succinate/fumarate and ETC as well as turn-over of acetyl-CoA and malonyl-CoA, building blocks for mycolic acids synthesis.

3) mmpL3 is implicated in TMM transport at the inner membrane. To have the full picture, is mmpL3 involved in the recycling of TMM/TDM in the process describe in this paper? This aspect could be discussed if not investigated here.

- 4) Metabolomics data should be deposited into a database as Metabolights (<https://www.ebi.ac.uk/metabolights/>).

- 5) The use of 10X MIC needs clarification. First, what is the MIC for the antibiotics for the strains used in this study? What is the rationale? The author should determine the MICs followed up by the determination of the MDK99 to have a clear understanding of the drug tolerance for the strains used.

- 6) The use of clinical isolates is an added value to the study but at this stage it seems important to measure the level of TMM/TDM in those strains as well for consistency with the proposed model and hypothesis.

- 7) Investigating the growth and phenotype/metabolomics of the mutant used in this study on acetate or propionate would have been relevant in the context of persisters in the host. This point could be discussed or potentially investigated by the authors.

- 8) Figure 2b, at day 28, only the $\Delta treS$ mutant exhibits a large increase in a lipid at the top of the TLC. What could be this lipid and why is it only present in the $\Delta treS$ mutant at day 28?

Point-by-point response

Reviewer #1

Manuscript "Transient drug-tolerance and permanent drug-resistance rely on the trehalose-catalytic shift in Mycobacterium tuberculosis"

This is an interesting, well-documented and comprehensive study on Mtb persisters generated during biofilm formation.

Thank you.

The authors use Mtb grown in biofilm as a model for drug tolerance: the biofilm matrix is enriched in what the authors call "persister-like bacilli (PLB)". To validate PLB as Mtb persisters, the authors compared the mRNA profiles of PLB at different time with D-cycloserine (DCS)-treated Mtb culture, based on published work that d-cycloserine generates persisters in Mtb. Metabolomes of 200 metabolites were followed in the PLB formation over the course of 35 days. The authors found that trehalose metabolism and central carbon metabolism pathways were among the most altered pathways. The authors deleted *treS*, the gene involved in maltose conversion from trehalose and found that $\Delta treS$ failed to form PLB. Although the paper is interesting and describes a new way to look at Mtb persisters, numerous items need to be addressed before it is suitable for publication.

See below.

Major Comments:

1) The authors state that $\Delta treS$ was deficient in persister formation in planktonic conditions using DCS, yet Fig. S5d does not seem to reflect this statement. The killing of $\Delta treS$ with DCS follows the same biphasic killing than H37Rv suggesting that persisters are indeed formed in this condition. Although $\Delta treS$ is more sensitive to DCS, the strain may not be deficient in persister formation in planktonic condition.

Thanks for the insightful comment.

Although we have found that $\Delta treS$ failed to form intact PLB similar to WT Mtb PLB in biofilm, our CV staining assay and time-course metabolomics (Fig. 2a and Fig. S5b) indicated that $\Delta treS$ can form PLB, albeit to a much lower extent and only after a significant delay. CV kinetics of WT Mtb and $\Delta treS$ PLB (Fig. S5b) showed that the formation of $\Delta treS$ PLB was initiated at day 28, which was much later as compared to day 16 for WT Mtb, indicating that Mtb evolutionarily developed multilayered adaptive strategies by which they cope with environmental stress (we added new text in Line 156). Thus, we don't exclude the possibility that $\Delta treS$ can form persisters in planktonic culture using DCS but if they do, it would be at a lower level and more slowly than WT.

As the reviewer pointed out, we repeated the DCS-persister formation experiment in planktonic culture conditions and reproduced a similar result (new graph in Fig. 5d). The CFU reduction pattern of $\Delta treS$ following DCS treatment showed biphasic kinetics similar to those of WT Mtb (Fig. S5d). However, the level of persister formation after DCS treatment in each time-segment (day 0 – 1; day 1 – 7; and day 7 – 14) was significantly greater in WT Mtb as compared to that of $\Delta treS$. The CFU reduction rate between days 0 – 1 was 94.1% in WT Mtb but 98.8% in $\Delta treS$; from days 1 – 7 it was 96.0% in WT Mtb but 99.1% in $\Delta treS$; and more importantly, between days 7 – 14 it was 70.6% in WT Mtb but 92.2% in $\Delta treS$. Collectively, WT Mtb following DCS treatment in planktonic culture produced 0.07% viable persister colonies while $\Delta treS$ produced

0.0008%, indicating that WT Mtb had an 87.5 fold-greater persister forming capability, likely due to an intact TreS-mediated trehalose-catalytic shift. As seen in biofilm culture, Δ treS still can form persisters in planktonic culture, but to a significantly lower extent. We added new text by clarifying that Mtb requires TreS activity to form persister-enriched population following DCS treatment and trehalose-catalytic shift is functionally active especially during days 7 – 14 (Lines 161 – 164).

We are confident that TreS deficiency destabilized not only PLB formation in biofilm but also persister formation following DCS treatment in planktonic culture.

2) The authors state that Δ treS may be more susceptible to ROS-mediated damage. This could be easily demonstrated by measuring CFUs/ml over time of Δ treS treated with hydrogen peroxide for example.

Thanks for the insightful suggestion.

As recommended, we added 2 mM H_2O_2 to WT Mtb and Δ treS in the presence or absence of 25 mM thiourea. As we speculated, we observed hyper-sensitivity of Δ treS against 2 mM H_2O_2 as compared to that of WT Mtb, due to a lack of a functional trehalose-catalytic shift. New text was added to Lines 227 – 228 and a figure was added as Fig. S7e.

3) Lane218-221: The concept that thiourea acts as an antioxidant in Mtb is controversial. To prove their point, the authors should include a killing curve with a known ROS producer in combination with thiourea.

Killing kinetics of WT Mtb, Δ treS, and Δ treS/com were monitored using a known ROS producer (H_2O_2) and the result was included in the response to the previous question. In addition to this, we also measured ROS production within WT Mtb and Δ treS following anti-TB drug (INH and BDQ) treatment using ROS fluorescence dye (dihydroethidium) and FACS. As seen in the following graph, there was a significantly higher accumulation of ROS within Δ treS as compared to that of WT Mtb, presumably due to a lack of a functional trehalose-catalytic shift. This provides the evidence that ROS induced by anti-TB drug is a main bactericidal effector, and that ROS-mediated damage is greater in Δ treS.

4) One of the authors conclusions is that the trehalose-catalytic shift serves as alternate biosynthetic sources of ATP, NADPH and antioxidants. A schematic of this alternate pathway should be added for clarification.

This has been added. Please see Fig. S12b.

Minor comments

1) Add unit to figure S6c.

We are sorry for our oversight. This has been corrected it in the revised manuscript.

2) Either add the concentrations of BDQ and CCCP used in Fig. S7 or indicate what the MIC values are.

This has been revised. We added 10X MIC equivalent concentration (0.3 µg/mL for BDQ and 20 µg/mL for CCCP).

Reviewer #2

The impressive study by Lee et al. shed lights the role of trehalose based lipids named TMM and TDM found in the cell envelope of Mycobacterium tuberculosis in the process of persisters bacilli and antimicrobial tolerance/resistance. This study is of high interest for the Tb community and can open new area of research of intracellular bacteria persisters.

The manuscript is clear, very-well written and the experiments have been conducted diligently with appropriate controls and validation. The authors used an impressive armoury of state-of-the-art methods to address the hypothesis.

Thank you.

However, at this stage, it seems required that the authors address those points:

1) It is know that lipid bodies and triacylglycerol are also involved in antimicrobial tolerance. Based on the actual manuscript, have the authors investigated the presence of lipid bodies in the PLBs in addition to metabolome remodelling towards trehalose synthesis?

Thanks for the insightful comment. Previous reports have shown that Mtb accumulated triacylglycerol (TAG) and free mycolic acids within bacteria when they enter the non-replicating state.

Mycolic acid accumulation must partly be associated with the trehalose-catalytic shift, in which trehalose is consumed for the biosynthesis of CCM (central carbon metabolism) intermediates while also kept away from the biosynthesis of TMM (trehalose + mycolic acid). Also, TAG accumulation has been shown to be mediated by *tgs1* (initial activity for TAG synthesis using acetyl CoA) activity. TAG accumulation in Mtb is achieved by activating Tgs1 function and thereby Tgs1 outcompetes CitA (TCA cycle enzyme that initiates TCA cycle intermediate biosynthesis by using acetyl CoA) in catalyzing acetyl CoA as a substrate, thus driving acetyl CoA towards the biosynthesis of TAG while reducing its flux towards TCA cycle intermediates. This may cause TAG accumulation in the non-replicating state and contribute to drug-tolerance.

As expected, we observed an accumulation of TAG and mycolic acids in PLB. Currently, we are studying the roles of TAG and mycolic acids in intrinsic drug-tolerance and regrowth using biofilm PLB.

2) To be consistent with the proposed hypothesis, as TMM and TDM required also fatty acid synthesis I and II, have the authors also consider investigating the turn-over of acetyl-CoA and Krebs cycle intermediates in their metabolomics data? This can potential bring interesting results regarding succinate/fumarate and ETC as well as turn-over of acetyl-CoA and malonyl-CoA, building blocks for mycolic acids synthesis.

The Mtb PLB characterization experiments (Fig. S1) indicated that PLB undergoes stresses as a result of ATP depletion that have been shown to occur as a consequence of TCA cycle remodeling (oxidative branch TCA cycle depletion) accompanied with NADH/NAD imbalance, ETC downregulation, and depletion of oxidative phosphorylation. Our metabolomics profiles of time-course PLB bacilli (Fig. S3c) also showed TCA cycle as a high-ranked pathway group that needs to be remodeled for PLB formation. Targeted metabolomics of PLB TCA cycle intermediates (data not shown) confirmed a significant reduction in both oxidative and reductive branch TCA cycle intermediates, obviating slowed turnover of acetyl CoA (presumably due to induced Tgs1 activity as explained in Reviewer2, question 1).

Although TCA cycle-mediated metabolic impact and drug-tolerance in Mtb are very interesting, this study was focused on the trehalose-catalytic shift and its accompanied metabolic roles. As mentioned, we are currently studying TCA cycle remodeling and mycolic acid metabolism in PLB using this same biofilm model.

3) *mmpL3* is implicated in TMM transport at the inner membrane. To have the full picture, is *mmpL3* involved in the recycling of TMM/TDM in the process describe in this paper? This aspect could be discussed if not investigated here.

We conducted qRT-PCR to monitor *mmpL3* expression during PLB formation and identified that *mmpL3* mRNA expression was downregulated throughout the course of PLB formation, suggesting a downregulation in TMM exporting activity. In recent reports (PMID: 30882198 and PMID: 22344175), Mary Jackson's group showed that the antimicrobial effects of MmpL3 inhibitors occur only on replicating Mtb, and not on hypoxic or non-replicating Mtb, thereby supporting our transcriptomics and metabolomics data of PLB. The MmpL3 activity is relatively dispensable in conducting trehalose-catalytic shift while forming PLB as compared to that in replicating state.

MmpL3 is known to be involved in exporting TMM synthesized inside the cytosol and serving a substrate for TDM biosynthesis. Thus, we speculated that the trehalose-catalytic shift in PLB is initiated either by enhancing TMM degradation or downregulating TMM biosynthesis; released trehalose is shunted to CCM.

New data has been added to Fig. S5a and associated text is included in the Result section (lines 174 – 177) and Discussion section (lines 323 – 327).

4) Metabolomics data should be deposit into database as Metabolights (<https://www.ebi.ac.uk/metabolights/>).

Once the manuscript is accepted, we will deposit the metabolomics raw data (time-course PLB of WT Mtb, $\Delta treS$, and $\Delta treS/com$, TB clinical isolates, and WT Mtb, $\Delta treS$, and $\Delta treS/com$ following treatment with BDQ or INH) into the nature webpage and Metabolights website.

5) The use of 10X MIC needs clarification. First, what is the MIC for the antibiotics for the strains used in this study? What is the rationale? The author should determine the MICs follow-up by the determination of the MDK99 to have a clear understanding of the drug tolerance for the strains used.

Against WT Mtb, we used 0.3 $\mu\text{g/mL}$ and 0.2 $\mu\text{g/mL}$ as 10X MIC concentrations of INH and BDQ, respectively. MIC was determined using a m7H9 liquid culture system. We exposed Mtb to 10X MIC bioequivalent doses of each drug against Mtb grown atop agar-supported filters at larger inocula suitable for filter-culture based system and metabolomics profiling platform.

As recommended, we monitored MDK99 (as described in PMID: 28636922) using 10X MIC of INH or BDQ. We employed the standard method of taking time-kill measurements following exposure to a certain concentration of antibiotics. The result identified that $\Delta treS$ showed almost half MDK99 (BDQ) time (~ 5 days) as compared to that of WT Mtb (~ 10 days) presumably due to a lack of trehalose catalytic shift. Conversely, MDK99 (INH) times between WT Mtb and $\Delta treS$ against INH treatment were similar (3.1 days, $\Delta treS$; 4 days, WT Mtb). The clear difference in MDK values between WT Mtb and $\Delta treS$ following treatment with BDQ is presumably due to a weak early bactericidal effect of BDQ against WT Mtb. The foregoing MDK determination affirmed that the presence of TreS confers stronger drug-tolerance of Mtb against BDQ.

6) The use of clinical isolates is an added value to the study but at this stage it seems important to measure the level of TMM/TDM in those strains as well for consistency with the proposed model and hypothesis.

Thanks for the insightful comment.

Consistent with metabolic profile of drug-sensitive and drug-resistant TB clinical isolates (Fig. S11d), we observed that trehalose supplementation enhanced the level of TMM/TDM only in drug-sensitive strains, but not in drug-resistant strains (especially in KT0384 and KT1111). Drug-resistant strains maintained or decreased the level of TMM/TDM in response to trehalose supplementation, supporting our metabolomics finding that trehalose-catalytic shift activity was greater in drug-resistant strains as compared to that of drug-sensitive strains. This is now included as Fig. S11e; related text has been added to the Results section (Lines 286 – 288 and 296 – 301).

7) Investigating the growth and phenotype/metabolomics of the mutant used in this study on acetate or propionate would have been relevant in the context of persisters in the host. This point could be discussed or potentially investigated by the authors.

Genetic and microbiological experiments implicated fatty acid as a potential carbon source encountered by Mtb during the chronic phase of infection. Here, we characterized our PLB (Fig. S1) and demonstrated that external carbon support was significantly decreased when PLB matured (at day 28 of biofilm), suggesting that trehalose-mediated metabolism in PLB is relatively unaffected by environmental carbon sources. We have added text in lines 363 – 364.

Indeed, during planktonic growth, $\Delta treS$ showed an almost identical growth pattern to that of WT Mtb in media containing acetate (Fig. S5c). This similar growth between WT Mtb and $\Delta treS$ cultured in media containing acetate was also confirmed by the finding that there were no major metabolic alterations on the $\Delta treS$ metabolome as compared to that of WT Mtb. This is visualized by a clustered heatmap (right panel) and a correlation plot (left panel) generated by metabolomics profile of WT Mtb and $\Delta treS$ cultured in acetate media.

Figure: The abundance of ~100 metabolites in WT Mtb, $\Delta treS$, and $\Delta treS/com$ cultured in media containing acetate for 24 hrs were analyzed and used to produce a correlation plot (left panel) and clustered heatmap (right panel). The correlation plot generated based upon the clustered heatmap showed high similarity (~1.0) between WT Mtb and $\Delta treS$ cultured in media containing acetate.

8) Figure 2b, at day 28, only the $\Delta treS$ mutant exhibits a large increase in a lipid at the top of the TLC. What could be this lipid and why is it only present in $\Delta treS$ mutant at day 28?

As shown in Fig. 1c, the top band was identified as free mycolic acids (confirmed by mycolic acid standard on the same TLC, lane 6 of Fig. 1c). We observed that this free mycolic acid was depleted in 28-day old PLB of WT Mtb whereas maintained in $\Delta treS$ PLB. It is presumably associated with the intrabacterial redox state affected by lack of trehalose-catalytic shift in $\Delta treS$ PLB. We are currently studying differences in mycolic acid metabolism between WT Mtb PLB and $\Delta treS$ PLB.

REVIEWERS' COMMENTS:

Reviewer #1 (Remarks to the Author):

The authors have carefully reviewed the critiques provided by the previous reviewers. They have addressed all points of concern and added sufficient information to support their studies.

Reviewer #2 (Remarks to the Author):

The authors addressed successfully all the comments and improved the manuscript.